# High Constitutive Cytokine Release by Primary Human Acute Myeloid Leukemia Cells Is Associated with a Specific Intercellular Communication Phenotype

**DOI:** 10.3390/jcm8070970

**Published:** 2019-07-04

**Authors:** Håkon Reikvam, Elise Aasebø, Annette K. Brenner, Sushma Bartaula-Brevik, Ida Sofie Grønningsæter, Rakel Brendsdal Forthun, Randi Hovland, Øystein Bruserud

**Affiliations:** 1Department of Clinical Science, University of Bergen, 5020,Bergen, Norway; 2Department of Medicine, Haukeland University Hospital, 5021 Bergen, Norway; 3Department of Medical Genetics, Haukeland University Hospital, 5021 Bergen, Norway; 4Institute of Biomedicine, University of Bergen, 5020 Bergen, Norway

**Keywords:** acute myeloid leukemia, gene mutations, differentiation, cytokines, proteomic profile, integrin, RAC1, SYK

## Abstract

Acute myeloid leukemia (AML) is a heterogeneous disease, and this heterogeneity includes the capacity of constitutive release of extracellular soluble mediators by AML cells. We investigated whether this capacity is associated with molecular genetic abnormalities, and we compared the proteomic profiles of AML cells with high and low release. AML cells were derived from 71 consecutive patients that showed an expected frequency of cytogenetic and molecular genetic abnormalities. The constitutive extracellular release of 34 soluble mediators (CCL and CXCL chemokines, interleukins, proteases, and protease regulators) was investigated for an unselected subset of 62 patients, and they could be classified into high/intermediate/low release subsets based on their general capacity of constitutive secretion. *FLT3*-ITD was more frequent among patients with high constitutive mediator release, but our present study showed no additional associations between the capacity of constitutive release and 53 other molecular genetic abnormalities. We compared the proteomic profiles of two contrasting patient subsets showing either generally high or low constitutive release. A network analysis among cells with high release levels demonstrated high expression of intracellular proteins interacting with integrins, RAC1, and SYK signaling. In contrast, cells with low release showed high expression of several transcriptional regulators. We conclude that AML cell capacity of constitutive mediator release is characterized by different expression of potential intracellular therapeutic targets.

## 1. Introduction

Acute myeloid leukemia (AML) is a heterogeneous hematological malignancy characterized by clonal proliferation of a hierarchically organized leukemia cell population that arises from hematopoietic progenitors in the bone marrow [1,2,3]. AML is distinguished from other related blood disorders by the presence of at least 20% myeloblasts in the bone marrow [1,2,3]. However, despite this common characteristic, AML is very heterogeneous [1], and patients differ, for example, with regard to genetic abnormalities [4,5,6,7], transcriptional [8] and cell cycle regulation [9], autocrine and paracrine growth regulation [10,11,12,13], as well as the cellular metabolomic [14] and proteomic profiles [15,16,17]. This cell population heterogeneity is also reflected in the biological characteristics of AML stem cells [8,10].

Most relapses occur within 2–3 years after diagnosis and the overall five-year leukemia-free survival for younger AML patients able to receive intensive chemotherapy possibly combined with stem cell transplantation is only 45–50%, and a major cause of death is chemoresistant AML relapse thought to originate from remaining AML or preleukemic cells that recapitulate disease development [18,19,20,21]. Cure is not possible for the large group of elderly/unfit patients who cannot receive such intensive therapy due to an unacceptable high risk of severe treatment-related morbidity or treatment-related mortality [2]. Thus, there is a need for identification of new therapeutic targets and development of new therapeutic strategies that are more efficient and better tolerated [22]. Targeting of the bidirectional communication between AML cells and their neighboring leukemia-supporting stromal cells is a possible approach [23,24,25,26,27,28]. In a previous study investigating another patient cohort, we described that high constitutive mediator release is associated with better long-term overall survival compared with low constitutive release [29]. The aims of the present study were, therefore, to characterize the in vitro secretome of primary human AML cells, to investigate possible associations between the capacity of constitutive mediator secretion and molecular genetic abnormalities, and to compare the proteomic profiles for primary AML cells with generally high and low capacity of releasing extracellular soluble mediators.

## 2. Materials and Methods

### 2.1. AML Patients and Preparation of Primary AML Cells

The study was approved by the Regional Ethics Committee (REK) (REK III 060.02, 10th of June 2002; REK Vest 215.03, 12th of March 04; REK III 231.06, 15th of March 2007; REK Vest 2013/634, 19th of March 2013; REK Vest 2015/1410, 19th of June 2015), The Norwegian Data Protection Authority 02/1118-5, 22 October 2002, and The Norwegian Ministry of Health 03/05340 HRA/ASD, 16 February 2004. All samples were collected after written informed consent.

The study population included 71 consecutive AML patients with high peripheral blood blast counts (>5 × 10^9^/L) and a high percentage of leukemic blasts among peripheral blood leukocytes (Table 1). Highly enriched AML cell populations (at least 95% leukemic blasts) could thereby be prepared by density gradient separation alone (Lymphoprep, Axis-Shield, Oslo, Norway). The cells were stored in liquid nitrogen until used in the experiments [30].

### 2.2. Mutation Profiling, Flow Cytometric Analyses, and Analysis of Global Gene Expression Profiles

Submicroscopic mutation profiling of 54 genes frequently mutated in AML was done by using the Illumina TruSight Myeloid Gene Panel and sequenced using the MiSeq system and reagent kit v3 (all from Illumina, San Diego, CA, USA). A detailed description of the methodology and the 54 genes is given in a previous publication [31]. Fragment analysis of *FLT3* exon 14–15, *NPM1* exon 12, and sequencing of *CEBPA* were performed as described previously [32].

Immunophenotyping was performed as a part of the standard diagnostic workup using freshly isolated cells [2], and analyses were performed by multiparametric flow cytometry (BD FACS Canto; Franklin Lakes, NJ, USA).

Our methods for analysis of global mRNA profiles have been described previously [31]. All these analyses were performed using the Illumina iScan Reader and based upon fluorescence detection of biotin-labeled cRNA. For each sample, 300 ng of total RNA was reversely transcribed, amplified, and biotin-16-UTP-labeled (Illumina TotalPrep RNA Amplification Kit; Applied Biosystems/Ambion; San Diego, CA, USA). The amount and quality of the biotin-labeled cRNA was controlled by the NanoDrop spectrophotometer and Agilent 2100 Bioanalyzer (Agilent Technologies, Inc.; Santa Clara, CA, USA). Biotin-labeled cRNA (750 ng) was hybridized to the HumanHT-12 V4 Expression BeadChip. The Human HT-12 V4 BeadChip targets 47,231 probes that are mainly derived from genes in the NCBI RefSeq database (Release 38). Data from the array scanning were investigated in GenomeStudio and J-Express 2012. All arrays within each experiment were quantile normalized before being compiled into an expression profile data matrix.

### 2.3. Analysis of Constitutive Mediator Release by Primary Human AML Cells

The studies of constitutive mediator release included a consecutive subset of 46 patients from the original study population (see Section 2.1 and Table 1). AML cells (1 × 10^6^/mL) were cultured for 48 h in Stem Span SFEM^TM^ medium in flat-bottomed 24-well (2 mL/well) culture plates (Nunc Micro-Well; Sigma-Aldrich, Saint-Louis, MO, USA) before supernatants were collected and stored at −80 °C until analyzed. The levels of the following 34 mediators were determined by Luminex analyses (R&D Systems; Minnesota, MN, USA) or enzyme-linked immunosorbent assays (ELISA) (R&D Systems; Minnesota, MN, USA): (i) the chemokines CCL2-5 and CXCL1/2/5/8/10/11; (ii) the interleukins IL-1β/1RA/6/10/33; (iii) the matrix metalloproteinases MMP-1/2/9 together with the protease/protease regulators tissue inhibitor of metalloproteinases 1 (TIMP-1), Cystatin B and C, polymorphonuclear (PMN) elastase, serpin C1 and E, and CD147, plasminogen activator (PA), and complement factor D (CFD); (iv) the immunomodulatory tumor necrosis factor-α (TNF); (v) the growth factors granulocyte-macrophage colony-stimulating factor (GM-CSF), hepatocyte growth factor (HGF), heparin-binding EGF-like growth factor (HB-EGF), basic fibroblast growth factor (bFGF), and vascular endothelial growth factor (VEGF); and (vi) the soluble angiopoietin-1 receptor tyrosine kinase with immunoglobulin-like and EGF-like domain 2 (Tie-2).

### 2.4. Proteomic Profiling: Selection of Patients, Sample Preparation, and Proteomic Analysis

The present study is based on mutational analysis of the leukemic cells for 71 consecutive and thereby unselected AML patients with a high number and/or percentage of AML blasts in the peripheral blood (Table 2). This selection based on the peripheral blood blast level (see Section 2.1) was used to reduce the risk of inducing molecular alterations in the leukemia cells due to more extensive separation procedures. The karyotyping (Table 1) as well as the mutational analyses showed an expected frequency of both cytogenetic and molecular genetic abnormalities, suggesting that despite the separation-dependent selection of patients, they are representative for AML in general. Constitutive cytokine release was investigated for a consecutive and thereby unselected subset of 46 patients from the original study population. Global proteomic profiling of enriched AML cells was performed for 16 of the 46 patients included in the constitutive release study; and these 16 patients represent all patients in the secretomic cohort completing intensive antileukemic treatment with induction chemotherapy followed by either 2–4 consolidation cycles or allogeneic stem cell transplantation as the final consolidation. Thus, they represent an unselected subset of relatively young and fit patients (Appendix A).

We followed the step-by-step procedure published previously for proteomic sample preparation and analysis of primary AML cells [15], except for the following two modifications: the 20 μg cell lysates were analyzed as label-free samples in contrast to being spiked with an internal standard, and no peptide fractionation was performed. The samples were analyzed on a QExactive HF Orbitrap mass spectrometer (Thermo Fisher Scientific; Waltham, MA, USA) coupled to an Ultimate 3000 Rapid Separation LC system (Thermo Fisher Scientific) [33,34]. The raw LC–MS files were searched against a concatenated reverse-decoy Swiss-Prot *Homo sapiens* fasta file (downloaded 05.03.18, containing 42,352 entries) in MaxQuant version 1.6.1.0 [35,36].

### 2.5. Bioinformatical and Statistical Analyses and Presentation of the Data

All statistical analyses were performed in GraphPad Prism 5 (GraphPad Software, Inc., San Diego, CA, USA). Unless otherwise stated, *p*-values <0.05 were regarded as statistically significant. The Fisher’s Exact test was used to compare different groups (two-tailed *p*-values). Bioinformatical analyses were performed using the J-Express 2009 analysis suite (MolMine AS, Bergen, Norway) [37]. Concentrations were then median normalized and transformed to logarithmic values before differences were analyzed. Unsupervised hierarchical clustering was performed with Euclidian correlation and complete distance measure for all analyses in J-Express. The Panther classification system (version PANTHER14.0) was used to identify distinct functional classes [38].

The proteomics data processing of the raw data (i.e., filtering for reverse hits, contaminants and proteins only identified by site, and log_2_ transformation of label-free quantification (LFQ) intensities), and statistical analysis of two groups using Welch’s *t*-test was performed in Perseus version 1.6.1.1. [39]. Furthermore, Z-statistics were used to find the proteins with the most abundant fold changes (FCs), i.e., the proteins with highest or lowest FC when comparing the high-release with the low-release group and calculating the FCs from the median log_2_ intensity per group as described by others [40]. Unsupervised hierarchical clustering was performed with Euclidian correlation and complete distance measure for all analyses in J-Express [37], and gene ontology analysis in DAVID version 6.8 [41]. Gene ontology (GO) terms with false discovery rate (FDR) < 0.05, the number of proteins associated to the term, and the fold enrichment were presented. The significantly different proteins were imported to the STRING database version 11.0 [42] to obtain protein–protein interaction networks, using experiments and databases as interaction sources at highest confidence (0.9). The networks were imported and visualized in Cytoscape version 3.3.0 [43]. Venny 2.1 (http://bioinfogp.cnb.csic.es/tools/venny/) was used to create Venn diagrams.

To summarize, due to the previously described AML heterogeneity and the fact that we sometimes have unequal numbers of quantified values of a protein in the two groups, we assumed an unequal variation in the groups and first applied the Welch *t*-test to identify proteins with significantly (*p* < 0.05) different mean tests. Thereafter we used Z-statistics as an additional test to identify those proteins with the most extreme/significant fold changes (fold change defined as the median intensity for high-release patients relative to the median intensity for low-release patients; the intensities were then log2-transformed).

## 3. Results

### 3.1. The Genetic Heterogeneity of AML Patients: TP53 Mutations are Associated with High-Risk Karyotypes and NPM1 Mutations are Associated with Mutations in DNA Methylation Genes

We analyzed the submicroscopic mutational profile for all 71 patients. The profile included 54 frequent mutated genes in myeloid malignancies, 37 of them carried non-benign mutations in our patients (Figure 1). At least one mutation was detected for 69 of the 71 patients, and one of patients without detected mutations had a balanced translocation. The median number of mutations per patient was 3.5 (range 0–7). The most frequently detected mutations were *NPM1* exon 12 insertion and the *FLT3*-ITD mutation (20 patients for each), followed by mutations in the *DNMT3A* (19), *TET2* (13), and *RUNX1* (13) genes (Appendix A).

We used the same (and now generally accepted) classification of AML-associated mutations in our present study as was used in two large previous studies, including 1540 and 200 patients, respectively [6,7]. The following mutations were detected in our patients: (i) *NPM1* insertion (detected in 20 out of the 71 patients), (ii) mutations causing activation of intracellular signaling (9 genes, 42 patients), (iii) mutated tumor suppressor genes (8 genes, 21 patients), (iv) mutations in genes involved in DNA methylation (5 genes, 39 patients) or (v) chromatin modification (3 genes, 15 patients), (vi) mutations in genes encoding myeloid transcription factors (3 genes, 20 patients), (vii) mutated genes important for the spliceosome (5 genes, 15 patients), (vii) mutated genes encoding cohesion proteins (3 genes, 9 patients), and (viii) the three genes *CSF3R*, *NOTCH1*, and *SETBP1* that were mutated in 5 patients (Table 2). The median number of different class mutations per patient was 2.5 (range 0–5); 24% of the patients had mutations from two different main classes and 34% from three main classes of mutations (Appendix A).

We compared the mutational status with karyotype, French–American–British (FAB) classification (i.e., morphological differentiation), de novo versus secondary leukemia, age, and gender (Figure 1); these statistical analyses are summarized in Appendix A. Firstly, we observed a highly significant association between *NPM1* and DNA methylation gene mutations (Fisher’s Exact test, *p* = 0.0015), whereas the association between *FLT3*-ITD and *NPM1* mutations did not reach significance. Secondly, there was a negative association between *NPM1* and myeloid transcription factor mutations (Fisher’s Exact test, *p* = 0.0001), and also between *NPM1* and chromatin modifier mutations that occurred together only for two patients. Thirdly, all patients with *TP53* mutations had high-risk cytogenetic abnormalities (Fisher’s Exact test, *p* < 0.0001). Fourthly, *NPM1* mutations were associated with morphological signs of differentiation, i.e., FAB classification M2/M4/M5/M6 (Fisher’s Exact test, *p* = 0.0233). Finally, even in this relatively small patient cohort, we observed that no patients with TET2 mutations (13 patients) had IDH mutation (5 patients); this inverse correlation has been described in previous cohorts [6], but did not reach statistical significance in our smaller cohort. We did not detect any significant associations between individual mutation or mutational main classes and age, gender, or AML etiology (de novo/secondary). A trend toward higher number of identified mutations in patients >65 years was detected, (median 4 mutations >65 years, and median 3 mutations <65 years), although did not reach statistical significance in this patient cohort. To summarize, the frequencies of individual mutations and the various associations are similar to what has been described previously [7,44]; the observations thus suggest that our patient cohort of consecutive patients with relatively high peripheral blood blast counts is representative for AML in general.

### 3.2. Expression of Molecular Differentiation Markers by Primary AML Cells: The Expression of the CD34 Stem Cell Markers Differs between Mutational Subsets

The AML cell expression of eight common differentiation markers (CD13, CD14, CD15, CD33, CD34, CD45, CD117, and HLA-DR) was available for 62 unselected AML patients. We first did an unsupervised hierarchical cluster analysis based on this expression profile (Appendix A). We could then identify four main patient subsets, but no single mutation or mutational class showed significant associations with any of the four main patient clusters.

We investigated whether there were any significant correlations between the CD34 stem cell marker and any of the other differentiation markers, but no significant associations were then detected.

We finally investigated whether any of the mutations that are used as prognostic markers in routine clinical practice [2] showed significant correlations with the expression of single differentiation markers. These statistical analyses are summarized in Appendix A. Firstly, *NPM1* mutations showed a significant correlation with CD33 expression (Fisher’s Exact test, *p* = 0.0107) and a negative association with CD34 expression (Fisher’s Exact test, *p* < 0.0001). These *NPM1* associations are similar to the observations in a previous large study of 184 unselected patients [45], and they are consistent with the observation that *NPM1* mutations are frequently associated with morphological signs of differentiation (see above). Secondly, neither *FLT3-*ITD nor *DNMT3A* mutations showed any association with CD34 expression. *NPM1* mutations are frequently combined with *FLT3*-ITD and DNA-methylation mutations [6], but only the negative *NPM1* association reached significance in our relatively small cohort. Thirdly, patients with mutations in chromatin modifier genes showed an increased frequency of CD34 expression by their AML cells (Fisher’s Exact test *p* = 0.0159). We detected the combination of *NPM1* and chromatin modifier mutations for only two patients, and this was similar to the observations in previous studies [7]. Thus, these mutational subsets also differ in their expression of differentiation markers, especially CD34 expression.

### 3.3. AML Patients Can Be Subclassified Based on Their Constitutive Release of Extracellular Mediators, but this Capacity Shows no Association with the Mutational Profile

Primary AML cells from 46 of the patients were available for additional studies of constitutive cytokine release during in vitro culture. This patient subset represents a constitutive and thereby unselected subset among the 71 patients included in our present study. We investigated the constitutive release of 34 soluble mediators, including several cytokines (interleukins, CCL and CXCL chemokines, immunoregulatory cytokines, growth factors), proteases, and protease regulators/inhibitors. A clustering analysis identified a subset of patients with generally high constitutive mediator release; the other patients showed generally low or intermediate release (Figure 2). Neither any single mutation nor mutational main class differed significantly when comparing the three patient subsets identified in this clustering analysis.

### 3.4. Comparison of Global Gene Expression Profiles for Patients with Generally High and Low Constitutive Release of Extracellular Mediators

We have previously described differences in global gene expression profiles between AML cells with generally high and low constitutive mediator release [46]. We performed a similar comparison for the patients included in the present studies based on the differentially expressed genes, and we could then identify two main patient subsets based on this expression (d-score >3.5; 149 genes identified). However, these two subsets did not differ significantly with regard to the distribution of single mutations or the overall mutational profiles of the AML cell populations (Appendix A).

### 3.5. Comparison of Proteomic Profiles for AML Cell Populations Showing Generally High and Low Constitutive Release of Extracellular Mediators

Our proteomic analyses identified 5852 proteins, but 5586 proteins were left after leaving out protein contaminants, reverse hits, and proteins only identified by site. Our further analyses were based on 4350 proteins that could be detected in at least five patients for each of the two compared groups. A significant difference (*p* < 0.05) in protein abundance between the two groups was detected for 256 of these proteins (182 proteins increased in patients showing high constitutive release, 74 proteins being increased in the others), i.e., determined by Welch’s *t*-test and Z-statistics (a list of selected proteins are described more in detail in Appendix A and the complete list of all 256 proteins is given in Appendix A).

We first performed an unsupervised hierarchical cluster analysis (Euclidean measure, and complete distance) based on the 256 differentially expressed proteins (Figure 3). Our analysis identified two main clusters/subsets of patients corresponding to patients with generally high and low constitutive release by their AML cells; only one of the high release patients clustered as an outlier. Furthermore, we performed GO term overrepresentation analyses based on the 256 differentially abundant proteins. The analysis of those proteins showing increased expression (*n* = 74) in patients with low constitutive mediator release and returned significantly increased GO terms, which reflected an altered regulation of nuclear functions/transcription/RNA metabolism (Table 3 and Appendix A). It can be seen that a major part of these genes are important for transcriptional regulation/RNA expression/RNA metabolism.

We then analyzed those proteins showing increased expression in AML cells with high constitutive cytokine release; the most significant GO-terms are listed in Table 4. When analyzing the proteins with regard to cell compartment the four largest terms (extracellular exosomes, cytosol, membrane, and cytoplasm) were only partly overlapping with regard to individual proteins and included 153 of the 182 proteins that were significantly increased in high-release AML cells (Figure 4). These four GO terms reflect cytoplasmic/cytosolic structures/functions together with the terms actin filament and phagocytic vesicle membrane. One of the terms reflects metabolic functions (NADPH oxidase complex), whereas the two last terms reflect cell surface functions/cellular communication (focal adhesion, membrane rafts). Analysis of biological processes and molecular functions included several relatively small GO terms that also reflect intracellular signaling, protein interactions, or cell surface receptor signaling (Table 4). Appendix A gives a more detailed description of those proteins that were identified both in the GO term analyses (Table 4, Figure 4) and in the network analysis (Figure 5; proteins in the large network to the left in the figure with increased levels in high-secreting cells).

The proteins with increased expression in patients with generally high constitutive release are presented in Figure 4 (all proteins included in the GO-terms GO:0070062—extracellular exosome, GO:0005829—cytosol, GO:0016020—membrane, or GO:0005737—cytoplasm); Table 3 (classification of proteins showing *p* < 0.01); Appendix A (description of proteins from Table 3 with *p* < 0.01); and Appendix A (the complete list of all 256 differentially expressed proteins). These more detailed analyses and classifications of individual proteins from Table 3 and Appendix A also show that AML cells showing generally high or low constitutive release of extracellular mediators differ especially with regard to transcriptional regulation, cell surface molecular profile, intracellular signaling, intracellular trafficking, and cell adhesion/migration.

We finally did a molecular network analysis based on the 256 differentially abundant proteins, and Figure 5 shows all molecular connections identified in this analysis (those molecules without any connections are left out). A total of 129 proteins were included in various networks; most of them appeared in a large network linked to the nodes spleen tyrosine kinase (SYK), NCF4 (a cytosolic regulator of superoxide-producing NADPH-oxidase), ARRB2 (regulator of G-protein-coupled receptor activity), ACTR3 (a major constituent of the ARP2/3 complex located at the cell surface and being essential for cell motility), and RAC1 (a GTPase belonging to the RAS superfamily of small GTP-binding proteins). Our overrepresentation analysis showed that exosomal proteins as well as proteins important for intracellular trafficking were differentially expressed; both these groups are important for communication from the leukemic cells to neighboring AML supporting stromal cells [47]. On the other hand, our network analysis showed that these AML cells had increased levels of several members of a signaling pathway, including cell surface integrins (αLβ2, αMβ2) known to mediate downstream signaling involving SYK and SRC kinase family members (FGR, HCK) [48,49,50,51]. Toll like receptor (TLR) 2 together with its downstream NFκB complex are also linked to this network [49]. Taken together these observations suggest that high constitutive extracellular release of soluble mediators is only a part of a more complex cellular phenotype that is characterized by differences in the bidirectional crosstalk between the leukemic cells and their neighboring AML-supporting cells. This bidirectional crosstalk involves cytokine-mediated signaling directed from the AML cells to the stromal cells. At the same time the stromal cells may influence the AML cells through soluble mediators or cell–cell contact with ligation of cell surface molecules, followed by downstream signaling (involving kinases and G-protein initiated signaling), and finally NFκB mediated modulation of cytokine/chemokine expression [48,49,50,51,52]. Finally, this crosstalk involves integrins that can mediate both inside–out and outside–in effects [48].

## 4. Discussion

AML is a heterogeneous disease, and this can also be seen from our present studies of primary human AML cells derived from a cohort of consecutive patients. In this study we focused on the molecular genetic abnormalities and the proteomic profiles of the leukemic cells [53]. Both the number and the nature of the molecular genetic abnormalities differed between the patients (number of detected mutations per patients 0–7, median 3.5 mutations). The frequencies of the various mutations were comparable to previous studies [6,7], *NPM1* mutations were associated with molecular and morphological signs of differentiation [45], and *TP53* mutations were associated with adverse karyotypes [54]. Taken together, these observations suggest that we investigated a representative AML patient population, even though we selected patients with relatively high peripheral blood blast counts/percentages.

In the present study, we included a group of consecutive and thereby unselected AML patients with a high percentage of leukemic blasts in peripheral blood. We used this selection of patients so that highly enriched AML cell populations could be prepared by density gradient separation alone; the risk of inducing molecular and/or functional alterations in the AML patients by more extensive cell separation procedures was thereby avoided [55]. Our results may therefore be representative only for this selected subset of patients, but several observations suggest that they possibly are representative for AML in general. Firstly, our patients showed an expected fraction of secondary versus de novo AML [56,57]. Secondly, as previously described in detail patients selected according to these criteria show a similar distribution of cytogenetic abnormalities as AML patients in general [30]. Thirdly, our present study shows that the distribution of various molecular genetic abnormalities is also similar to AML in general [6,7,44,58]. Finally, we have described in detail the selection of the 16 AML patients included in our proteomic studies (see Section 3.5), and they should then be regarded as representative for relatively young AML patients.

Extensive separation procedures will influence the functional characteristics of primary human AML cells, and one would expect that in vitro incubation in culture medium would have similar effects. However, previous studies have shown that the characteristics of even long-term cultured primary human AML cells are associated with patient survival [59], an observation suggesting that even in vitro cultured cells will reflect functional characteristics of clinical relevance.

Distinct immunophenotype profiles may be associated with specific mutations, and search for immunophenotype-based screening approaches have therefore been suggested [60,61]. We investigated the immunophenotype profiles of individual patients based on the expression of eight differentiation markers commonly used for classification of myeloid cells. We identified four different main clusters/patient subsets based on this profiling, but no single mutation or mutation main classes showed significant associations to any of these profiles. However, associations between mutations and single differentiation markers were observed, especially expression of the CD34 stem cell marker that was negatively associated with *NPM1* mutations as well as *FLT3*-ITD and *DNMT3* mutations, whereas chromatin modifier mutations were positively associated with CD34 expression. Such associations have also been described previously [45,62]. A possible explanation for this is that single mutations may have a major impact on the expression of single or related markers, whereas the overall mutational profile has a major impact on the overall differentiation profile.

In previous studies we showed that the constitutive release of a wide range of soluble mediators by primary AML cells varied considerably between patients, and a subset of patients then showed a generally high release compared with other patients that either showed intermediate or low release [11,52]. This capacity of constitutive mediator release was tested in a highly standardized in vitro model. We investigated the constitutive release for a consecutive subset of our patients, and again we found that a subset of patients showed generally higher release of most mediators compared with the other patients. We then selected those samples that were derived before the first time of diagnosis for all relatively young patients that completed intensive chemotherapy. We compared the proteomic profiles of the primary AML cells for eight patients showing high and another group of eight patients showing generally lower mediator release.

Several proteins were differentially expressed when comparing patients with generally high and low constitutive cytokine release. The high release patients showed high expression, especially of proteins involved in intracellular signaling, intracellular transport/trafficking and communication between cells (soluble mediators, exosomes, cell surface molecules, and intracellular mediators downstream to cell surface receptors). We did not identify any of the soluble mediators when analyzing differentially abundant cell proteins between the two patient subsets; this is not unexpected because there is often not a strong correlation between cellular levels and extracellular release of soluble mediators during culture [34].

The high constitutive mediator release should in our opinion be regarded as only a part of a more complex communication phenotype with neighboring non-leukemic stromal cells. In contrast, the cell populations with low constitutive release showed increased abundance proteins involved in or regulating gene transcription/RNA synthesis/RNA metabolism. A possible hypothesis may be that cells with high constitutive release have a higher dependency on neighboring AML-supporting stromal cells than leukemia cells showing low constitutive release. We would emphasize that primary AML cells have a wide range of secreted biomolecules, which can be useful in classification/prognostication and as therapeutic targets [11,52,63]. The mediators included in the present study are well-characterized and are released at detectable levels for most patients. For these reasons they should be regarded as biologically important in the disease, but they probably represent only a part of the AML cell secretome that is involved in the bidirectional crosstalk between leukemic and non-leukemic cells in their common bone marrow microenvironment.

## 5. Conclusions

We conclude that the high constitutive extracellular release of soluble mediators by primary human AML cells seems to reflect a complex functional phenotype with regard to communication between AML cells and their neighboring non-leukemic stromal cells in their common bone marrow microenvironment. Our proteomic comparison has identified high expression in this patient subset of several intracellular molecules that are regarded as possible therapeutic targets in human AML. Dual targeting of intracellular signaling and extracellular communication should therefore be considered for these patients.

## Figures and Tables

**Figure 1 jcm-08-00970-f001:**
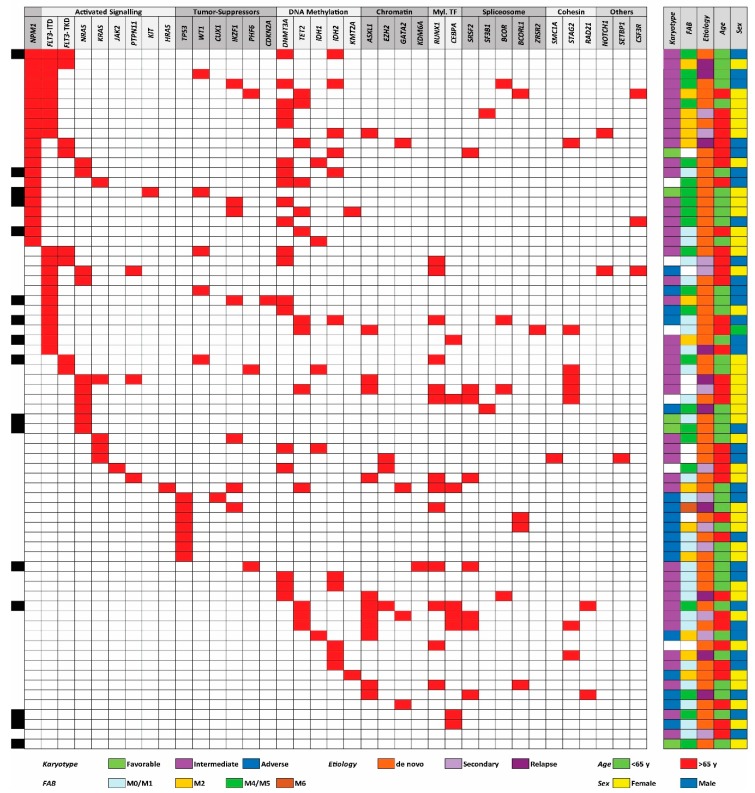
The total genomic profile and organization of mutations into defined categories; an overview of the data for the 71 AML patients included in our study. The figure shows the somatic mutations identified from a 54 gene mutation panel, the mutations being classified as described previously [6,7]. A majority of 69 patients had at least one detectable mutation. Risk classification of the karyotypes, morphological signs of differentiation (i.e., FAB-classification), etiology, age, and gender are presented in the right part of the figure. The patients selected for proteomic analyses are indexed with black in the left part of the figure.

**Figure 2 jcm-08-00970-f002:**
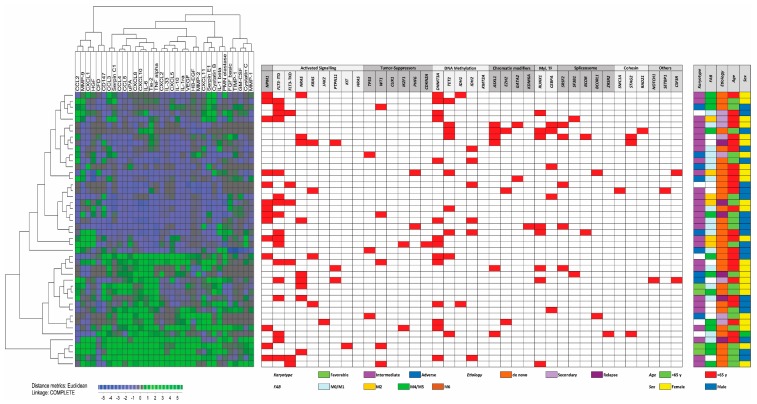
The secretome and genomic profile for 46 AML patients. Primary AML cells derived from a consecutive subset of 46 patients were cultured in vitro for 48 h and the supernatant levels of 34 soluble mediators were then determined. We performed an unsupervised hierarchical cluster analysis (Euclidean measure, and complete distance) based on these results and were then able to identify two distinct patient clusters corresponding to patients with generally high or intermediate/low supernatant level.

**Figure 3 jcm-08-00970-f003:**
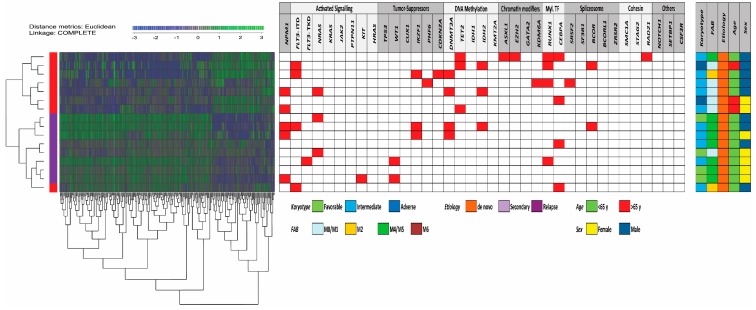
Identification of two main patient subsets based on proteomic differences of AML cells with high and low constitutive release. Eight of the 16 patients included in the proteomic studies belonged to the cluster characterized by generally high constitutive mediator release and the eight others showed low/intermediate secretion (Figure 2); 256 proteins differed significantly between these two groups. We performed an unsupervised hierarchical cluster analyses (Euclidean measure, and complete distance) based on the levels of these proteins, and the left part demonstrates the dendrogram and heat map; blue indicates low protein levels and green high levels. Two main clusters were then identified corresponding to the high and low/intermediate secretion patients except for one outlier patient (left column, red color indicating high release). As expected, the two main clusters were heterogeneous with regard to mutational frequencies (middle panel) and did not differ with regard to clinical or biological characteristics either (right panel).

**Figure 4 jcm-08-00970-f004:**
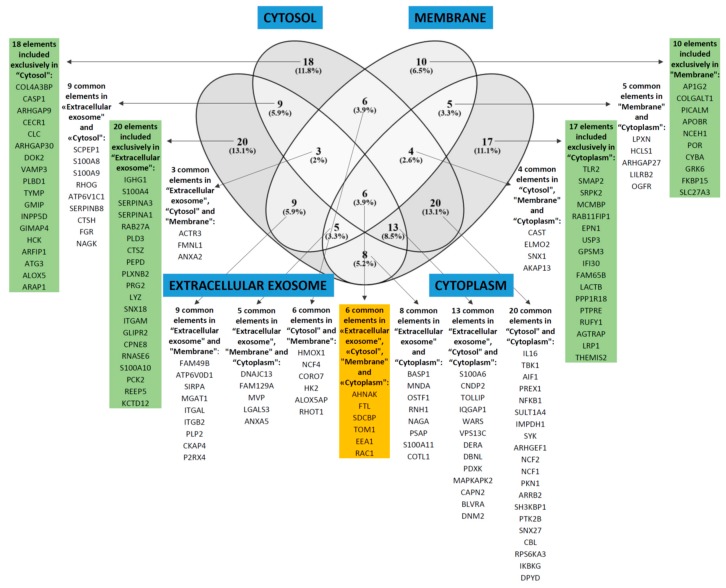
GO-terms including significantly increased proteins for AML cells with generally high constitutive release of extracellular soluble mediators. The over-representation analysis based on cellular compartment identified four GO terms with FDR < 0.05 and including at least 40 proteins, i.e., GO:0070062—extracellular exosome, GO:0005829—cytosol, GO:0016020—membrane, and GO:0005737—cytoplasm. These four GO-terms were partly overlapping (only six proteins included in all four); together they included 153 of the 186 proteins that were increased in AML cells with generally high constitutive release compared with AML cells with low/intermediate constitutive release.

**Figure 5 jcm-08-00970-f005:**
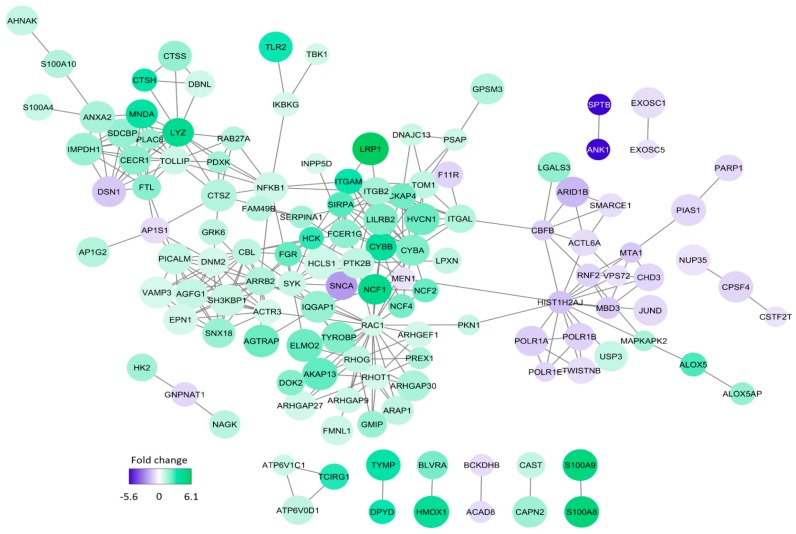
The network analysis of proteins showing differential expression in primary AML cells with generally high versus generally low constitutive release of extracellular mediators. The intensity of the color reflects the fold change (FC) significance when comparing the high- and low-release groups; thus a negative fold change indicates increased protein abundance in the low-release group (purple) and a positive fold change indicates increased protein abundance in the high-release group (green). This STRING-DB analysis was based only on the 256 proteins that were quantified and considered significantly different between the two groups; the figure thus shows proteins from our quantified data and no shells of interactors were considered.

**Table 1 jcm-08-00970-t001:** The clinical and biological characteristics of the 71 acute myeloid leukemia (AML) patients included in the study.

Age and gender		Etiology	
Median (years)	64	Previous chemo-radiotherapy	1
Range (years)	18–90	CML	1
Females	31	Li–Fraumeni’s syndrome	1
Males	40	Polycythemia vera	1
		MDS	8
		Relapse	10
		de novo	49
FAB^1^ classification		Cytogenetic abnormalities^3^	
M0/1	26	Adverse	17
M2	14	Favorable	5
M4/5	22	Intermediate	43
M6	1	Normal	40^4^
Unknown	8	Unknown	6
CD34 expression			
Negative (<20%)	28^2^		
Positive (>20%)	43		

^1^ The French–American–British classification. ^2^ The percentage of positive cells in flow cytometric analysis. ^3^ The European Leukemia Net classification was used [2]. ^4^ The 43 patients classified as intermediate cytogenetics included 40 patients with normal karyotype. Abbreviations: CML, chronic myeloid leukemia; MDS, myelodysplastic syndrome.

**Table 2 jcm-08-00970-t002:** An overview of the mutational landscape of 71 consecutive AML patients. The table presents the main classification and the number of mutations. For each main class the term total group refers to the total number of mutations in this class (first number) together with the number of patients with mutations belonging to this main class (second number). Those mutations that should be included as a part of the prognostic evaluation in routine clinical practice are marked with arrows (↑ increased survival; ↓ decreased survival) [2].

Classification	Mutation	Number with Mutation	Classification	Mutation	Number with Mutation
**NPM1**	↑NPM1	20	**Chromatin modification**	↓ASXL1	12
	Total group	20–20		EZH2	3
**Signaling**	↓FLT3-ITD	20		GATA2	4
	FLT3-TKD	8		KDM6A	1
	HRAS	1		Total group	20–15
	JAK2	1	**Myeloid transcription factors**		
	KIT	1		↑CEBPA	8
	KRAS	5		↓RUNX1	13
	NRAS	10		Total group	21–18
	PTPN11	3	**Spliceosome/transcription repressors**	BCOR	4
	Total group	49–42		BCORL1	4
**Tumor suppressors**	CDKN2A	1		SF3B1	2
	CUX1	1		SRSF2	8
	IKZF1	7		ZRSB2	1
	PHF6	3		Total group	19–15
	TP53↓	7	**Cohesin**	RAD21	2
	WT1	5		SMC1A	1
	Total group	24–21		STAG2	8
**DNA methylation**	DNMT3A	19		Total group	11–11
	IDH1	5	**Others**	CSF3R	3
	IDH2	11		NOTCH1	2
	KMT2A/MLL	2		SETBP1	1
	TET2	12		Total group	6–5
	Total group	49–39			

**Table 3 jcm-08-00970-t003:** Differentially expressed proteins in primary AML cell populations with high (left) and low (high) constitutive release of extracellular soluble mediators. The mediators are classified based on their main functional characteristics. The information is based on the Gene database and selected references from the PubMed database (Appendix A). The proteins being increased in high-secreting AML cells are those proteins that were both included in the gene ontology (GO) terms GO:0070062—extracellular exosome, GO:0005829—cytosol, GO:0016020—membrane, and GO:0005737—cytoplasm (Figure 4), and also in the main interacting protein network in the left part of Figure 5 (Appendix A). The proteins being increased in the low-secreting AML cells are those proteins included the GO terms GO:0000790—nuclear chromatin and GO:0005736—DNA-directed RNA polymerase I complex (Table 4).

Main Classification	Increased Protein Levels in Cells with High Constitutive Release	Increased Protein Levels in Cells with Low Constitutive Release
**Nucleosome**		MBD3
**Chromatin, histone, transcription, RNA**	TOLLIP, NFKB1	HIF0, HISTIH2AJ, MTA1, SMARCE1, MEN1, MBD3, POLR1E, CLPX, POLR1A, POLR1B
**DNA repair**		CLPX, JUND, POG2
**Oncogene**	CBL, DBNL	
**Cell cycle regulation**	IL16	
**Intracellular signaling**	SYK, HCLS1, AKAP1, TLR2, TOLLIP, AGTRAP, ANXA2, CECR1, INPP5D, LPKN, IKBKB, TBK1	
**Tyrosine kinase**	SYK, HCLS1, FGR, PKN1	
**SRC tyrosine kinases**	HCLS1, FGR, HCK,	
**PI3K-Akt-mTOR**	NCF4	
**RAC1**	RAC1, NCF4, RHOT1, ARHGEF1, PKN1, RHOG, ARHGAP30, PREX1, GMIP, DOK2, AKAP1	
**GTPase**	DNM2, ARHGEF1, PKN1, RHOG, ARHGAP30, PREX1, GMIP, AKAP1, ARHGAP, RAB27A	
**G-protein coupled receptors**	ARRB2, ARHGEF1, PREX1, GRK6	
**Phagocytosis**	CYBA, NCF2, NCF4, ELMO2	
**Protein degradation**	CBL, SERPINA1	
**Intracellular trafficking**	VAMP3, DNM2, PICALM, SNX18, ARAP1, ARAP1, TOLLIP, AP1G2, S100A10, S100A4, TOM1, SDCDP, DNAJC13, EPN1, APHGAP, RAB27A	
**Microtubule, cytoskeleton, structure**	DNM2, EPN1, SH3KBP1, PKN1, RHOG, AHNAK, SDCDP, S100A4, CKAP4, FAM49B	
**Cell migration**	PLXNB2, HCK, DNM2, RHOG, ELMO2, AHNAK	
**Mitochondria, metabolism**	FAM49B, FTL, IMPDH1, PDXK	CLPX
**Lysosomes**	CTSH, CTSS, CTSZ, LYZ, PSAP	
**Cell metabolism, NADP**	HCK, NCF4	
**Cytokinesis**	FMNL1	
**Extracellular matrix, cell adhesion**	EPN1, SH3KBP1	
**Extracellular mediators**	IL16, TLR2, TOLLIP	
**Cell surface molecules**	ITGAL, ITGAM. ITGB2, SYK, LILRB2, PKN1, LPXN	
**Integrins**	ITGAL, ITGAM. ITGB2, SYK, FGR, LPXN	
**Viability, apoptosis**	SH3KBP1, PKN1, ARAP1, TLR2	
**AML**	CBL, PICALM	
**Differentiation**	MNDA, NCF1, CECR1	

**Table 4 jcm-08-00970-t004:** Significant GO-terms (i.e., FDR < 0.05) for proteins showing significantly increased levels in AML patients with intermediate/low and high constitutive mediator release.

**Low constitutive mediator release; list of significant GO-terms**		**Protein number**	**Fold enrichment**	**FDR**
**Cell compartment**	GO:0005654—nucleoplasm	31	2.8	2.3 × 10^–5^
	GO:0000790—nuclear chromatin	8	11	0.0099
	GO:0005736—DNA-directed RNA polymerase I complex	4	80	0.017
**Molecular function**	GO:0003713—transcription coactivator activity	9	8.5	0.011
	GO:0001054—RNA polymerase I activity	4	78	0.018
**High constitutive mediator release; list of significant GO-terms**				
**Biological processes**	GO:0006954—inflammatory response	19	5.0	6.5 × 10^–5^
	GO:0045087—innate immune response	20	4.7	8.3 × 10^–5^
	GO:0048010—vascular endothelial growth factor receptor signaling pathway	9	13	8.6 × 10^–4^
	GO:0007229—integrin-mediated signaling pathway	10	10	9.4 × 10^–4^
	GO:0031623—receptor internalization	7	16	0.0062
	GO:0007165—signal transduction	29	2.5	0.015
	GO:0098609—cell–cell adhesion	13	4.8	0.026
**Cell compartment**	GO:0070062—extracellular exosome	73	2.7	1.4 × 10^–13^
	GO:0005829—cytosol	79	2.5	5.7 × 10^–13^
	GO:0016020—membrane	48	2.3	7.8 × 10^–5^
	GO:0043020—NADPH oxidase complex	5	43	0.0048
	GO:0005737—cytoplasm	78	1.6	0.010
	GO:0030670—phagocytic vesicle membrane	7	12	0.026
	GO:0005925—focal adhesion	15	4.0	0.03
	GO:0045121—membrane raft	11	5.6	0.038
	GO:0005884—actin filament	7	11	0.046
**Molecular function**	GO:0005515—protein binding	129	1.4	5.8 × 10^–6^
	GO:0017124—SH3 domain binding	11	8.9	5.8 × 10^–4^
	GO:0035325—Toll-like receptor binding	4	96	0.0058

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
