# Peer review of "High Constitutive Cytokine Release by Primary Human Acute Myeloid Leukemia Cells Is Associated with a Specific Intercellular Communication Phenotype"

_jcm, 2019, doi:10.3390/jcm8070970_

Reviewer 1 Report

The authors present a well written manuscript on their results regarding a genetic and proteomic Analysis of Primary AML cell sample for 71 patients treated in Norway.

Overall the paper should be published but undergo major revision before.

I would like to raise some comments and needs for clarification here: 

Line 45: Please specify the time point the survival rate of 45%-50% relates to. Is this a 5-year survival probability?

Line 52: In the introduction you gave a rather broad overview on AML and Mention briefly (as a motivation) that Analysis of "bidirectional communication" between blasts and stromal cells is a "possible approach". I would strongly suggest to put your research into context of already published analyses. There will be at least so articles describing Research on blasts signalling that should cited and set into relation to your Analysis in the introduction and also in the discussion. 

Line 119: You very generally state that Chi-Squared Tests were used to compare Groups. Some Questions arise when reading the results. First of all the Chi-Squared test has one assumption About the expected frequency in the cross-table cells. If this is not met your test will be unreliable and Fisher's exact test should be used. Please check all(!) your analyses if this might be the case. Second: Your extensive testing strategy has a high risk of false-positive results. While within the proteomic analysis FDR control was used for the GO Annotation (probably because ist directly provided within Cytoscape) you should consider all(!) of you Chi-Square Tests to be part of a multiple testing problem and consider FDR Control also for the other results. 

Please then extent your description of the statistical analysis accordingly.

Line 123: Since the Panther System is versioned it would be important to report the date of the analysis and the Panther version that was active then. 

Line 127: Please further motivate the usage of Z-statistics and t-Tests in the proteomics analysis. 
Do you mean the test statistic of the t-test, or are these two complimentary testing approachs to find some association?

Line 131: Same comment for STRING

Line 132: How many "Shells" in the STRING-DB Analysis were considered?

Line 133: Please cite Venni 2.1 Software.

Line 122 and 129: Please also add information about how clusters were identified after generating the dendrogram. 

Line 144: Please briefly explain in the methods section what you published in [8]. Not clear for the reader.

LIne 156: Duplication of "(first number)". Parts of the table are broken to the next page. 

Line 159 / Figure 1: Color coding is suboptimal for some variables. E.g. Karytypoe Groups are hard to distinguish. Adverse karytype and "selected for proteomics" have the same Colour. Grey 

Background in colum headings makes it hard to read.

Line 169 (and all occurences of test results): Please report an overview of all Tests as supplementary tables. Within the main text of the manuscript please also report also a measure/Information on effect size beside the p-value. This could be the Chi-Squared teststatistics or any Information on relative frequencies in the Groups. Consider FDR control for the whole paper (see above).

Line 176: There is evidence that the mutational load grows with Patient age. You state that there is no association between individual mutations and classes with age. This may be true but you could also check for the number of mutations here. 

Line 184: how were the 4 subgroubs identified. See also comment in Methods section. 

Line 218 / Figure 2: There seems to be a missmatch in rows betwen the left side heatmap and the Mutation table. Please check! In the figure caption "consecutive" patients is written. Is this still true for the 46 patients. 

Line 219: Eucledian distances were used but the Methods section says "spearman correlations"

Line 231: Consider moving the whole Paragraph to the method section. 

Line 283: "(next page)" in Figure caption.

Line 291: Not clear what is meant by "Bonferroni and Benjamini-corrected" since only FDR is reported. Please defined Fold Change in the material and Methods section. 

Line 298/Table 4: Some Splitting Information on the table is written in the middle which probably is not necessary. 

Line 310: "(previous page)" comment seems unnecessary. 

Line 312: "signfiicance" means FDR corrected p-values?

Figrue 5: Is it possible to add Information on the function categories (GO Terms) to the Network diagram? The darkest nodes are not readble to the black font on the Background Color.

Line 444: Döhner et al 2017 is duplicated. Please delete and renumber all subsequent references also in the main text. 

Figure S2: Left side heat map: What are the Grey cells? No data? How was this considered in the cluster Analysis?

Table S2: Both Groups are labeled "high release". 

Table S3: Not clear if the Levels are medians of log2 transformed Expression measurement (why not log10?) or log2 transforms of the median Expressions values?

The Ratio is the difference on the log scale. Please be clear on your definitions here. 

General: Are you using the z-score (which is a standard normal quantiles) or the p-Value?? Because a low zscore indicates no-difference! If this is a p-value (which you denote as z.score) this is incorrect and you should say this is a z.test p-values (which the z.score beloings to as z.statistics). 

Author Response

The authors present a well written manuscript on their results regarding a genetic and proteomic Analysis of Primary AML cell sample for 71 patients treated in Norway.Overall the paper should be published but undergo major revision before.I would like to raise some comments and needs for clarification here.

1.1 Line 45: Please specify the time point the survival rate of 45%-50% relates to. Is this a 5-year survival probability?

This has now been rewritten as suggested by the reviewer (page 2, line 44).

 1.2 Line 52: In the introduction you gave a rather broad overview on AML and Mention briefly (as a motivation) that Analysis of "bidirectional communication" between blasts and stromal cells is a "possible approach". I would strongly suggest to put your research into context of already published analyses. There will be at least so articles describing Research on blasts signalling that should cited and set into relation to your Analysis in the introduction and also in the discussion. 

 We have added six new references. They are all review articles that describe the communication between AML cells and their neighboring stromal cells, including effects of receptor ligation on intracellular signaling. It is not possible to cover all forms of intercellular signaling and intracellular signaling, but these references cover several aspects of the crosstalk between leukemic and stromal cells (including the stem cell niche cells) in the bone marrow. We hope our selection of review articles is acceptable.

 1.3 Line 119: You very generally state that Chi-Squared Tests were used to compare Groups. Some Questions arise when reading the results. First of all the Chi-Squared test has one assumption About the expected frequency in the cross-table cells. If this is not met your test will be unreliable and Fisher's exact test should be used. Please check all(!) your analyses if this might be the case. Second: Your extensive testing strategy has a high risk of false-positive results. While within the proteomic analysis FDR control was used for the GO Annotation (probably because ist directly provided within Cytoscape) you should consider all(!) of you Chi-Square Tests to be part of a multiple testing problem and consider FDR Control also for the other results. 

 The Fischer’s Exact test has now been used throughout the manuscript instead of the Chi-Square test. This is explained in the Material and methods section (page 4, line 152).

The use of the test and the problem of multiple testing are now addressed in the Results section, and an additional table is included in the supplementary information (Table S3). 

 We agree that multiple testing has to be addressed, for this reason we have made a comment in Table S3 where we indicate which p-values that were significant also after Bonferroni corrections. We emphasize that we did not compare everything with everything; we did a limited number of statistical analysis to verify that our patient population is representative and shows the expected correlations/associations previously described in larger patient populations. Our present cohort is too small to detect new associations, and our main message is the proteomic profiling the associations between protein levels and the ability of constitutive extracellular mediator release.

 1.4 Please then extent your description of the statistical analysis accordingly.

 Please see the response to comment 1.3.

 1.5 Line 123: Since the Panther System is versioned it would be important to report the date of the analysis and the Panther version that was active then. 

 The requested information is now included t in the article (page 4, line 157).

 1.6 Line 127: Please further motivate the usage of Z-statistics and t-Tests in the proteomics analysis. Do you mean the test statistic of the t-test, or are these two complimentary testing approaches to find some association?

 We apologize for not making this clear to the reader. We used the Welch t-test, which assumes un-equal variation in the groups, to test whether the mean intensity of each protein was equal or not in the two groups. We chose this test and not a Student’s t-test (which assumes equal variance) due to the well-known large heterogeneity of AML patients and AML cells derived from different patients (this is illustrated also by the information given in our present study), and because we sometimes have unequal numbers of quantified values of a protein in each group (i.e. sometimes a protein has been quantified in all eight patients in one group but only in five patients in the other group. This is often due to peptide fragmentation in the mass spectrometry analysis, which a relatively random event. Furthermore, we used Z-statistics to capture the proteins with more extreme fold changes, because we assume that they will be the most interesting from a biological point of view. The fold changes determined from the median high-release intensity versus the median low-release intensity was used for this calculation, and the intensities are log2-transformed for all calculations. In this test, the percentiles are used to calculate the upper and lower standard deviations of the distribution of protein fold changes, and the test returns the tails of the distribution (i.e. the more extreme fold changes in both directions). This test is further explained in the reference we have cited: Arntzen MØ et al., IsobariQ: Software for Isobaric Quantitative Proteomics using IPTL, iTRAQ, and TMT. Journal of Proteome Research, 2011 (Reference 40 in the revised version of the article).

 Thus, we first apply the Welch t-test to find proteins with a significantly (p<0.05) different mean (in this study 409 of 4350 proteins). Thereafter we use the Z-statistics as an additional test to find the proteins with the most extreme fold changes, and 256 of the 409 proteins with a significant p-value also had a significant fold change.

 We have extended the description of our statistical methods in the Revised Version of our manuscript (page 4, line 161). Furthermore, Z-statistics were used to find the proteins with the most significant fold changes (FC) (i.e. the proteins with highest or lowest FC when comparing the high-release group to the low-release group) and this was calculated from the median log2 intensity per group, as previously described by others. To clarify this better for the readers we have added a summarizing chapter at the end of section 2.5.

 1.7 Line 131: Same comment for STRING

 This has been corrected in the revised manuscript. String version 11.0 was used (page 4, line 168).

 1.8 Line 132: How many "Shells" in the STRING-DB Analysis were considered?

 Our STRING-DB analysis was only based on the 256 proteins we quantified and considered significantly different between the groups. Thus, Figure 5 only shows proteins from our quantified data and no shells of interactors were considered. This is now stated in the figure legend.

 1.9 Line 133: Please cite Venni 2.1 Software.

 This has now been included (page 4, line 171).

 1.10 Line 122 and 129: Please also add information about how clusters were identified after generating the dendrogram. 

 The choice of clusters was based on visualization of the dendogram made by the Euclidean distance measure between the different identified clusters. This is explained in the revised version (page 4, line 155).

 1.11 Line 144: Please briefly explain in the methods section what you published in [8]. Not clear for the reader.

 This is a large study describing the genomic and epigenomic landscapes in AML based on studies of 200 patients (new reference 7). Another study included 1540 patients and described the genomic classification and prognosis in AML (new reference 6). We have carefully rewritten this sentence to briefly describe these studies to the readers (page 5, line 190).

 1.12 Line 156: Duplication of "(first number)". Parts of the table are broken to the next page. 

 This has now been corrected.

 1.13 Line 159 / Figure 1: Color coding is suboptimal for some variables. E.g. karyotype groups are hard to distinguish. Adverse karyotype and "selected for proteomics" have the same color. Grey background in column headings makes it hard to read.

 The figure has been redesigned as suggested by the reviewer (page 6). We hope our solutions are acceptable.

 1.14 Line 169 (and all occurrences of test results): Please report an overview of all Tests as supplementary tables. Within the main text of the manuscript please also report also a measure/Information on effect size beside the p-value. This could be the Chi-Squared teststatistics or any Information on relative frequencies in the Groups. Consider FDR control for the whole paper (see above).

Please see our response to comment 1.3.

1.15 Line 176: There is evidence that the mutational load grows with Patient age. You state that there is no association between individual mutations and classes with age. This may be true but you could also check for the number of mutations here. 

 A trend towards more detected mutations in older patients was detected; median 3 in patients<65 4="" years="" and="" median="" in="" patients="">65 years. However, in this material it did not reach significance. We have comment this  in our revised manuscript (page 7, line 230)

 1.16 Line 184: how were the 4 subgroubs identified. See also comment in Methods section. 

 The methods for making dendograms and clustering identification are currently described (page 4, line 155).

 1.17 Line 218 / Figure 2: There seems to be a mismatch in rows between the left side heatmap and the Mutation table. Please check! In the figure caption "consecutive" patients is written. Is this still true for the 46 patients. 

 The figure has now been corrected. The text has not been altered; it is still true that 46 patients were included and these patients are consecutive.

 1.18 Line 219: Eucledian distances were used but the Methods section says "spearman correlations"

 This is corrected in the revised version

 1.19 Line 231: Consider moving the whole Paragraph to the method section. 

 We have moved this section to the Material and methods section and is now a part of Section 2.4. We have rewritten the heading of this section to reflect this. We have also included references to other relevant sections and tables in this chapter; a short additional comment is also added to section 2.3.

 1.20 Line 283: "(next page)" in Figure caption.

 This has been deleted.

 1.21 Line 291: Not clear what is meant by "Bonferroni and Benjamini-corrected" since only FDR is reported. Please defined Fold Change in the material and Methods section. 

 We are grateful for this comment. In our study we have presented GO terms having a FDR < 0.05, and in our original version we stated in this table that also the Bonferroni and Benjamini-corrected p-values were < 0.05. Since FDR is the strictest correction, an FDR < 0.05 necessitates that Bonferroni and Benjamini-corrected p-values also are<0.05. We therefore removed the parenthesis stating “all terms also showed Bonferroni and Benjamini-corrected p-values <0.05” from the table legend, and simply state that added “GO terms with false discovery rate (FDR) < 0.05 are presented” (legend, Table 3).

 1.22 Line 298/Table 4: Some Splitting Information on the table is written in the middle which probably is not necessary. 

 This has been removed.
 1.23 Line 310: "(previous page)" comment seems unnecessary. 

 The comment has been removed.

 1.24 Line 312: "significance" means FDR corrected p-values?

 This refers to the fold change analysis; this is now stated in the figure legend (Figure 6).

 1.25 Figure 5: Is it possible to add Information on the function categories (GO Terms) to the Network diagram? The darkest nodes are not readable to the black font on the Background Color.

 This is a good idea, but we were not able to make a nice and easily interpretable figure when including the GO terms. We therefore prepared a figure using the ClueGO application available through Cytoscape; but in our opinion this figure did not add anything to the presentation we therefore left it out.

 Regarding Figure 5: We have changed the black font to improve readability (we also had to increase the node size slightly). We added a color key to better explain what the colors of the nodes indicate (see lower left part).

 As mentioned it turned out to be very difficult to make a figure that integrated the network analysis and the GO terms. Instead we have extended Table S4 (previous Table S3) to include all proteins that were included in the most important GO terms (for the low-release patients) and all proteins that were included in the four main GO terms (see Figure 4) and at the same time were included in the main interacting network to the left in Figure 5 (i.e. proteins with (i) increased levels in high-release patients, (ii) included in this large interacting protein network to the left in Figure 5, and (iii) also included in the four main GO terms stated in Table 3).

 We hope this last solution is acceptable. Although it not as easy to read as a common figure would have been, but at least more detailed information about all the proteins that are identified in both bioinformatical analyses is now easily available to the reader. We believe that this will also be a help for the reader.

1.26 Line 444: Döhner et al 2017 is duplicated. Please delete and renumber all subsequent references also in the main text. 

 The reference list has been corrected, the original references 2 and 4 were identical.

 1.27 Figure S2: Left side heat map: What are the Grey cells? No data? How was this considered in the cluster Analysis?

 This is missing data, and is currently comment in the figure legend to Figure S2. The cluster analysis was performed as standard algorithms taken into account missing data.

 1.28 Table S2: Both Groups are labeled "high release". 

 We are grateful for this comment. The mistake has been corrected and we have changed the “High release” to the right to “Low release”.

 1.29 Table S3: Not clear if the Levels are medians of log2 transformed Expression measurement (why not log10?) or log2 transforms of the median Expressions values?

 In the methods section we state “For the proteomics data processing of the raw data (i.e. filtering for reverse hits, contaminants and proteins only identified by site, and log2 transformation of LFQ intensities)” (Page 4, Line 159). All further calculations are based on these numbers. It is stated in the table legend of the present Table S4 “the median protein levels (log2 transformed) for AML cells with generally low and high constitutive release of extracellular mediators” and for clarity we have changed it to: the median protein levels (of log2 transformed protein intensities) for AML cells with generally low and high constitutive release of extracellular mediators”.

 Both log2 and log10 could have been used; both are commonly used and we have chosen log2 transformation. Both are commonly used in bioinformatical analyses when using expression values and they should give the same results.

 1.30 The Ratio is the difference on the log scale. Please be clear on your definitions here. 

 First of all, we have changed the term “ratio” to “fold change”, since this is the term we use throughout the manuscript. The fold change is calculated from the medians (already in log2, since the individual intensities are log2 transformed). Using HIF0 (the first protein in Table S4) as example: Fold change (-2.40) = level high release (25.30) – level low release (27.70).

 The fold changes are presented in log2 scale. We have stated this clearly in the heading of the present Table S5 and we have also changed the legend of the present Table S4 to:

 “The results are presented as the protein identity (presented by gene name) together with the Welch’s t-test p-value, the median protein levels (of log2 transformed protein intensities) for AML cells with generally low and high constitutive release of extracellular mediators, and the fold change (in log2 scale) for each mediator when comparing the level of high constitutive AML cell release cells versus low release cells.”

 1.31 General: Are you using the z-score (which is a standard normal quantiles) or the p-Value?? Because a low z score indicates no-difference! If this is a p-value (which you denote as z.score) this is incorrect and you should say this is a z.test p-values (which the z.score belongs to as z.statistics). 

 We would like to thank the reviewer for suggesting a correct way of conveying the results. We have changed the table legend and table heading of the present Table S5 accordingly: “the two right columns show the z-test p-value and the Welch’s t-test p-value”.

Reviewer 2 Report

The article by  Reikvam H et al mainly collected more than 70 AML samples, analyzed their mutations, expression of molecular differentiation markers,  characterized their in vitro secretome. Their discovery could provide critical scientific hints for clinical therapy through disturbing the crosstalk AML cells with high constitutive release of extracellular mediator and their neighboring non-leukemic stromal cells. The study is interesting and timely. However, before publication, the authors should revise the manuscript.

(1) Line 93, Page 3: the authors showed that "AML cells were cultured for 48 hours....". So what kind of AML cells indicated here? Bone marrow mononuclear cells, peripheral mononuclear cells, CD34+ cells, or others? The authors should clearly show which kind of AML cells in every experiment.

(2) Fig 1. In this part, the authors characterized the mutation status of mutation in AML patients they utilized, and concluded their discovery is similar to what was described previously. In AMLs, it seems that two different mutations usually exclusive with each other, such as TET2 mutation and IDH mutation. Did they authors also observe the exclusive mutation in their samples?

(3) Results 3.2 The authors tried to present whether there exists the correlation between CD34 stem markers and differentiation markers, or between CD34 expression and gene mutations. It would be much better if the authors could plot a heatmap including r values and p value from each analysis. Thus the readers can get your conclusion in an easier way.

(4) How about the correlation between the patients with high constitutive release of extracellular mediators and overall survival? 

Author Response

The article by Reikvam H et al mainly collected more than 70 AML samples, analyzed their mutations, expression of molecular differentiation markers,  characterized their in vitro secretome. Their discovery could provide critical scientific hints for clinical therapy through disturbing the crosstalk AML cells with high constitutive release of extracellular mediator and their neighboring non-leukemic stromal cells. The study is interesting and timely. However, before publication, the authors should revise the manuscript.

 3.1 Line 93, Page 3: the authors showed that "AML cells were cultured for 48 hours....". So what kind of AML cells indicated here? Bone marrow mononuclear cells, peripheral mononuclear cells, CD34+ cells, or others? The authors should clearly show which kind of AML cells in every experiment.

 All experiments were based on the use of gradient-separated highly enriched mononuclear cells from the peripheral blood. This is now clearly stated in the Material and methods section (page2, line 69).

 3.2 Fig 1. In this part, the authors characterized the mutation status of mutation in AML patients they utilized, and concluded their discovery is similar to what was described previously. In AMLs, it seems that two different mutations usually exclusive with each other, such as TET2 mutation and IDH mutation. Did they authors also observe the exclusive mutation in their samples?

 This was also observed in our samples, as can be seen from our figure. Although no TET2 positive samples had IDH mutations our present cohort is too small for a statistical analysis as this exclusion was demonstrated in larger patient cohorts. We have note comment this further in our revised version

 (3) Results 3.2 The authors tried to present whether there exists the correlation between CD34 stem markers and differentiation markers, or between CD34 expression and gene mutations. It would be much better if the authors could plot a heatmap including r values and p value from each analysis. Thus the readers can get your conclusion in an easier way.

 Our statistical analyses were also commented by reviewer 1 who wanted us to use the Fisher’s exact test for these statistical analyses (see comment reviewer 1.3). We have therefore done this, and we have also prepared a separate supplementary table (Table S3) where we summarize all the analyses. This Table will also make it easy for the reader to see the results from our analyses and function as an overview. We hope this solution is acceptable; even though it is not a heat map it is a helpful illustration for the reader and an overview of the analyses.

(4) How about the correlation between the patients with high constitutive release of extracellular mediators and overall survival

 The association between the survival after intensive chemotherapy was described in another publication where we investigated another and larger patient cohort. This is now stated in the Introduction and the reference is given (page 2, line 53, reference 29). However, the patients were selected according to the same criteria as the patients in the present article (consecutive and thereby unselected, circulating blasts allowing preparation of highly enriched leukemia cell population by gradient separation alone without more extensive cell separation methods that could induce functional alterations in the AML cell populations).

 Reviewer 3 Report

Håkon Reikvam et al. analyzed the relationship between released cytokines and genetic mutations, compared the proteomic profiles of two contrasting patient subsets, concluded that the AML cells capacity of constitutive mediator release is characterized by different expression of potential intracellular therapeutic targets. The manuscript was organized well, and the data are abundant and convincing. Still, there are some issues needed to be addressed before considering publishing on our Journal, here are my points:

 1. Did authors deposit the raw array data into any public space? We encourage authors to do this and provide accessibility to our readership.

2. The labels in Fig1, FigS2 right panel and Fig2 right panel are too small to see clearly.

3. Genetic mutations are always associated with poor/favorite diagnosis, please comment in the 3.2 paragraph, together with the cell surface markers to discuss the possible relationship.

4. In vitro culture in Stem Span was totally different from in vivo microenvironment, this limitation should be mentioned in the discussion.

Author Response

Håkon Reikvam et al. analyzed the relationship between released cytokines and genetic mutations, compared the proteomic profiles of two contrasting patient subsets, concluded that the AML cells capacity of constitutive mediator release is characterized by different expression of potential intracellular therapeutic targets. The manuscript was organized well, and the data are abundant and convincing. Still, there are some issues needed to be addressed before considering publishing on our Journal, here are my points.

 2.1 Did authors deposit the raw array data into any public space? We encourage authors to do this and provide accessibility to our readership.

 A raw data of the microarray/mRNA data will be made available as suggested in Gene Expression Omnibus (GEO).

 2.2. The labels in Fig1, FigS2 right panel and Fig2 right panel are too small to see clearly.

 These figures have been corrected.

 2.3 Genetic mutations are always associated with poor/favorite diagnosis, please comment in the 3.2 paragraph, together with the cell surface markers to discuss the possible relationship.

 We have now indicated the prognostic impact in Table 2 for those molecular genetic abnormalities that should be included in the prognostic evaluation of AML patients in routine clinical practice. This explained in the heading of the table and in the text (section 3.2), and a reference is given both in the table legend and in the text.

 2.4. In vitro culture in Stem Span was totally different from in vivo microenvironment, this limitation should be mentioned in the discussion.

 A new chapter addressing this question is now included in the Discussion section (page 25, line 414).